# Quadrature squeezing in a nanophotonic microresonator

Alexander E. Ulanov[1] ✉, Bastian Ruhnke[1], Thibault Wildi[1] & Tobias Herr [1,2] ✉

Squeezed states of light are essential for emerging quantum technology in metrology and information processing. Chip-integrated photonics offers a route to scalable and efficient squeezed light generation, however, parasitic nonlinear processes and optical losses remain significant challenges. Here, we demonstrate single-mode quadrature squeezing in a photonic crystal microresonator via degenerate dual-pump spontaneous four-wave mixing. Implemented in a scalable, low-loss silicon-nitride photonic-chip platform, the microresonator features a tailored nano-corrugation that modifies its resonances to suppress parasitic nonlinear processes. In this way, we achieve an estimated 7.8 dB of on-chip squeezing in the bus waveguide, with potential for further improvement. These results open a promising pathway toward integrated squeezed light sources for quantum-enhanced interferometry, Gaussian boson sampling, coherent Ising machines, and universal quantum computing.

Squeezed light is an essential resource for emerging quantum technologies, including gravitational wave detection[1], quantum metrology[2], imaging[3], and quantum computing[4,5]. It can be generated through nonlinear parametric processes and in one of its quadratures exhibits noise below that of classical coherent laser sources[6–8]. Since the seminal demonstration of 0.3 dB noise reduction in hot sodium vapor in 1985[9], the field has made significant progress, culminating in the current record of 15 dB achieved using a periodically poled KTP crystal cavity[10]. In many systems, the squeezing process can be accompanied by competing nonlinear effects that can obstruct the observation of squeezing and hinder its practical utilization. Thus, to enable effective generation of squeezed light, it is crucial to not only optimize the squeezing process itself but also to suppress unwanted nonlinear processes. Moreover, it is important to minimize optical losses as they degrade the level of squeezing.

Complementing squeezed light sources based on bulk optics, significant advances have been made in the development of photonic chip-integrated sources[11] based on spontaneous parametric downconversion in $\chi^{(2)}$ nonlinear waveguides, such as periodically poled thin-film lithium niobate[12–17], and four-wave mixing in $\chi^{(3)}$ Kerr-nonlinear materials[18–25].

Kerr-nonlinear systems are attractive for scalable quantum technologies as they provide access to ultra-low loss photonic-integrated waveguides and high-quality factor $Q$ optical microresonators ($Q > 10$ million)[26–29], enabling efficient excitation of nonlinear effects and squeezing. Importantly, they can also be produced in existing wafer-scale processes and are compatible with hybrid integration of laser sources and detectors.

However, $\chi^{(3)}$ Kerr-nonlinear systems exhibit rich four-wave mixing dynamics, with multiple nonlinear processes occurring simultaneously[30,31]. If not suppressed, some of these processes can compromise the level of squeezing, especially for the generation of single-mode squeezed vacuum (SMSV) – a key resource for continuous-variable quantum information processing[32]. Therefore, suppressing unwanted nonlinear processes remains a key challenge in such systems. To address this, the pump lasers can be detuned from the respective microresonator modes[22,31]. While this mitigates parasitic effects, it reduces the achievable level of squeezing for a given pump power. A more power-efficient approach involves the use of linearly[23,33] and nonlinearly[34] coupled microresonators, where careful thermal tuning of both resonators can suppress unwanted nonlinear processes and lead to efficient squeezing. Sharing many similarities with coupled microresonators, nano-corrugated photonic crystal rings (PhCR) have

[1]Deutsches Elektronen-Synchrotron DESY, Hamburg, Germany. [2]Physics Department, University of Hamburg UHH, Hamburg, Germany. ✉e-mail: alexander.ulanov@desy.de; tobias.herr@desy.de

emerged as a new paradigm in integrated nonlinear photonics. In a PhCR forward and backward propagating waves can be coupled in a precisely controlled manner (Fig. 1a). They have enabled significant progress in soliton microcombs[35–37], slow light[38], optical parametric oscillators[39], and broadband control of microresonator dispersion[40].

Here, we demonstrate efficient single-mode quadrature squeezing based on a dual-pumped spontaneous four-wave mixing process in a single Kerr-nonlinear PhCR microresonator. Critically, the coupling induced by the PhCR's nano-corrugation modifies the resonance spectrum such that unwanted nonlinear optical processes are effectively suppressed, resulting in an estimated 7.8 dB of on-chip squeezing. Compared to approaches based on coupling of independent resonators, the spectral modification induced by the nano-corrugation is static, thus making the system immune to environmental or pump-induced temperature variation – a favorable property for large-scale application. Moreover, the resonator is fabricated via UV lithography in a commercial foundry process and hence compatible with scalable wafer-level fabrication, as needed for emerging quantum technologies.

## Results

Our squeezed light generator is based on dual-pump spontaneous four-wave mixing (DP-SFWM), where two pump lasers provide parametric gain for the signal field, located spectrally between the two pumps (mode S, see Fig. 1c). Below the threshold of optical parametric oscillation, this process results in SMSV generation. However, DP-SFWM is accompanied by other nonlinear effects, including cross-phase modulation (XPM), self-phase modulation (SPM), single-pump spontaneous four-wave mixing (SP-SFWM), and Bragg-scattering four-wave mixing (BS-FWM) (Fig. 1c). While XPM and SPM induce frequency shifts that can be easily compensated for by small detunings of the pump lasers, SP-SFWM and BS-FWM introduce excess noise into the signal mode (S) if not suppressed. Therefore, optimal SMSV generation in these systems requires effective suppression of SP-SFWM and BS-FWM processes. As we explain below, we leverage a single PhCR microresonator to achieve the suppression of these processes.

The general concept of a PhCR is illustrated in Fig. 1a. A spatial periodic corrugation along the resonators inner sidewall with angular corrugation period $\theta_0 = \pi/m_0$, creates a controllable coupling with rate $\gamma$ between the counter-propagating optical waves of angular (azimuthal) mode number $m_0$. The coupling of the degenerate forward-backward propagating modes creates a set hybridized modes with a frequency splitting of $2\gamma$. In our case, to suppress SP-SFWM and BS-FWM processes, we utilize a single PhCR with a periodic corrugation pattern composed of two Fourier components, designed to achieve

strong splitting of modes $X_{\pm2}$ (Fig. 1c, d), so that the unwanted processes are no longer supported by the resonances. This is similar to the engineering of photonic stop bands in waveguides for photon pair generation[30].

To design the PhCR, as a first step, we analyze the impact of parasitic nonlinear channels on the achievable squeezing level and determine the resonance splitting required to mitigate these effects. To this end, we consider a system comprising five adjacent resonances of a high-$Q$-factor microresonator. The system is pumped by two monochromatic lasers aligned with the $P_{\pm1}$ resonances, respectively (Fig. 1c). We assume that the pumps are undepleted, the intracavity fields reach a stationary state, and all other modes remain below the oscillation threshold. Under these assumptions, the system dynamics is described by a set of coupled-mode equations governing the evolution of the creation and annihilation operators[41]. By incorporating cavity input-output relations[42], we derive a model for the squeezing spectrum (Methods). This framework can be generalized to accommodate an arbitrary number of modes and pump fields, allowing for the identification of maximally squeezed supermodes[43,44].

The simulated squeezing and anti-squeezing spectra, i.e., minimal and maximal quadrature variance (over all quadrature phase angles) in dependence of offset frequency $\Omega$, are shown in Fig. 2a for two pumps with equal powers and identical detunings from their respective resonances. The simulations were conducted for three different values of the normalized splitting $\beta = 2\gamma/\kappa$ of the resonances $X_{\pm2}$, where $\kappa = \kappa_0 + \kappa_{ex}$ is the resonator's total loss rate, combining its intrinsic loss rate $\kappa_0$ and its coupling rate $\kappa_{ex}$ to the bus waveguide. For each value of $\beta$, the pump powers and detunings are adjusted to ensure that the system remains just below the oscillation threshold. As illustrated in Fig. 2a, the antisqueezing level remains nearly constant across the simulations, while the squeezing level progressively improves with increasing $\beta$, eventually converging at low offset frequencies $\Omega$ to the maximally attainable squeezing (deterministic bound) for a specific cavity outcoupling efficiency $\eta = \kappa_{ex}/\kappa$ (here we assume $\eta = 0.9$). These results show that splitting the parasitic resonances effectively mitigates the detrimental effects of SP-SFWM and BS-FWM nonlinear processes. In this model, a splitting of $\beta \gtrsim 15$ is sufficient to nearly reach the deterministic bound of 10 dB. The $\beta$-dependence of the squeezing level is further detailed in the Supplementary Information (SI), Sec. 3.

For the experiments, we fabricate a range of over-coupled resonators with varying corrugation amplitudes. At the mask design level, the corrugation amplitudes for both Fourier components are set to be equal, ensuring approximately equal splitting of the target resonances. The microresonators are fabricated using a commercial foundry

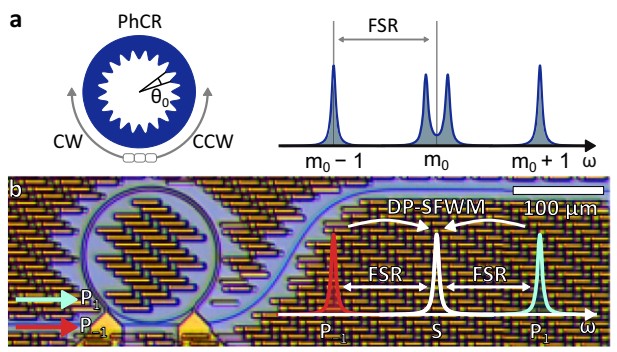

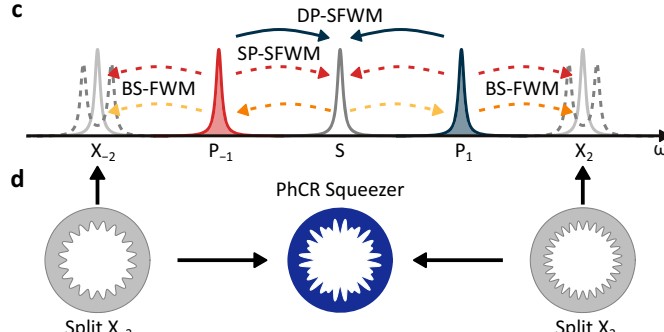

**Fig. 1 | Dual-pump spontaneous four-wave mixing (DP-SFWM) in a photonic crystal ring microresonator (PhCR). a** A PhCR with a periodic corrugation pattern, which induces strong coupling between counter-propagating optical waves resulting in frequency splitting of mode $m_0$ (angular period $\theta_0 = 2\pi/2m_0$). **b** Two pump fields aligned with resonances $P_{\pm1}$ provide parametric gain for the mode $S$, situated between the two pumps. **c** Resonance structure and nonlinear mixing

processes induced by two monochromatic pumps aligned with resonances $P_{\pm1}$, including dual-pump spontaneous four-wave mixing, single-pump spontaneous four-wave mixing (SP-SFWM), and Bragg-scattering four-wave mixing (BS-FWM). **d** PhCR squeezer with the corrugation composed of two Fourier components that are aligned with modes $X_{\pm2}$, resulting in strong splitting of these modes, effectively suppressing SP-SFWM and BS-FWM processes.

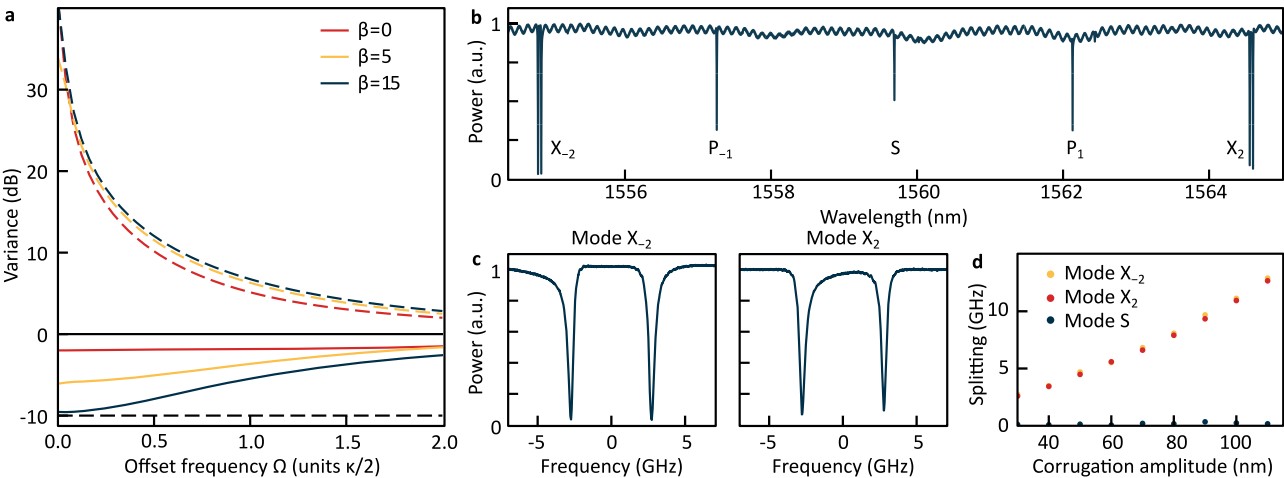

**Fig. 2 | Simulation of squeezing spectra and characterization of photonic crystal ring microresonator. a** Numerical simulation of shot-noise normalized squeezing and antisqueezing spectra. The red, orange, and blue lines correspond to different values of splitting of resonances $X_{\pm 2}$ (Fig. 1c). The black dashed line indicates the deterministic bound, determined by the cavity outcoupling efficiency, here assumed as $\eta = 0.9$. **b** Transmission trace obtained during a laser wavelength scan across multiple resonances of the photonic crystal ring resonator under study. **c** Zoom-in view of two split resonances $X_{\pm 2}$, showing significant frequency splitting due to the corrugation pattern aligned with these modes. Other resonances remain unaffected. **d** Measured frequency splitting $2\gamma$ for the two split resonances $X_{\pm 2}$ and the $S$ resonance (represented by red, orange, and blue dots, respectively) as a function of the designed corrugation amplitude.

process (see "Methods") and have a free spectral range (FSR) of 300 GHz, corresponding to a radius of ~75 µm. The waveguides have cross-sectional dimensions (W × H) of $1.6 \times 0.8\ \mu m^2$, resulting in micro-resonators with anomalous group velocity dispersion $D_2/(2\pi) \approx 10$ MHz. The microresonator is coupled to a straight bus waveguide via a point-type evanescent coupler, and coupling to the bus waveguide is accomplished via edge-coupling at the uncoated facets of the chip through inverse tapered spot-size converters.

We characterize the fabricated resonators using frequency comb-calibrated laser scans[45], enabling the measurement of the coupling rate $\gamma$ between counter-propagating modes, the intrinsic loss rate $\kappa_0$, and the cavity coupling rate $\kappa_{ex}$. The samples exhibit an intrinsic loss rate $\kappa_0 \approx 2\pi \cdot 55$ MHz, corresponding to an unloaded quality factor $Q_0$~3.5 million at a signal wavelength of ~1559 nm. The cavity outcoupling efficiency is $\eta \approx 0.9$, close to the value assumed in the simulations presented above. We expect that through the use of pulley couplers it could potentially be increased to ~0.95.

A resonator transmission spectrum is presented in Fig. 2b. It shows that only the two targeted resonances (corresponding to the modes $X_{\pm 2}$ in Fig. 1c) exhibit significant splitting (Fig. 2c), while the S mode remains unaffected. Here, we select a sample with frequency splitting of ~5.4 GHz ($\beta \approx 13.5$), which is sufficient to suppress parasitic nonlinear processes effectively (SI, Section 3). Figure 2d shows that such large dual mode splitting can be achieved controllably and without noticeable increase in loss, consistent with previous work[36].

A schematic of the experimental setup for generating and measuring squeezing is shown in Fig. 3a. The frequency- and phase-locked dual-pump is derived from a single tunable continuous-wave (CW) master laser using electro-optic modulation, followed by amplification and filtering (SI, Section 1). The pump components are adjusted to have equal power before being coupled into the chip via a lensed fiber. The output light is collected using a large numerical aperture asphe-rical lens and directed to a volume Bragg grating filter, which transmits the residual dual-pump light while reflecting the generated squeezed light. The reflected light, after being overlapped with the local oscil-lator (LO), is directed to a free-space balanced homodyne detector (BHD). The LO is derived from the master laser to ensure coherence and frequency match with the generated squeezed light. The output signal from the BHD is then analyzed using an electronic spectrum analyzer (ESA).

To operate the setup, the electro-optic modulation frequency is adjusted to match the frequency separation of the pump modes and the CW laser is tuned so that the pumps approach their respective resonances from the blue side (alternatively, the resonator may be thermally tuned to match the pump frequency). Frequency tuning of the CW laser is halted when both pumps are slightly blue-detuned relative to the effective resonance frequencies, ensuring relative detuning stabilization via thermal locking[46]. This approach eliminates the need for active frequency stabilization.

A characteristic shot-noise normalized quadrature variance trace measured in this state at 20 MHz sideband frequency as the LO phase is swept is shown in Fig. 3b. The directly measured squeezing (anti-squeezing) levels are 1.71 dB (5.54 dB), respectively. From these values, the total efficiency is inferred to be $T \approx 0.37$ (see "Methods"), con-sistent with the independently measured total efficiency of $T \approx 0.40$ obtained by multiplying the efficiencies of individual components (Methods). The remaining discrepancy is attributed to other unquan-tified imperfections, such as optical losses on the BHD windows, sub-optimal light collection, incomplete signal overlap with the photodiodes, and unsuppressed parasitic nonlinear processes (SI, Section 3). The level of squeezing depends on the pump power. This is evidenced in Fig. 3c by the measured squeezing and anti-squeezing levels at a sideband frequency of 20 MHz and further detailed in the SI, Section 4.

Finally, we characterize the squeezing spectrum. For this purpose, we fix the total pump power at 33 mW, below the parametric threshold of ~40 mW (SI, Section 5) and sweep the offset frequency $\Omega$ (on the ESA in zero-span mode) and the LO phase simultaneously, to trace out the (anti-)squeezing spectra as minima and maxima of the obtained curve. The results, are shown in Fig. 3d, where we have accounted for the frequency dependence of the BHD detection (efficiency and dark noise clearance (DNC); SI, Section 2 and "Methods"), to reproduce the actual shape of the squeezing spectrum. At low offset frequencies $\Omega$, we achieve 2.4 dB squeezing and 8.3 dB anti-squeezing. Considering the respective experimental uncertainties, these levels correspond to $11.3 \pm 0.7$ dB of squeezing in the cavity and $7.8 \pm 1.1$ dB in the bus waveguide at 90% outcoupling efficiency (1$\sigma$ confidence intervals). At 400 MHz offset frequency, the levels of squeezing and antisqueezing are 0.85 dB and 1.3 dB, respectively. The squeezing bandwidth is lim-ited by the cavity linewidth and, as the red and yellow traces in Fig. 3d

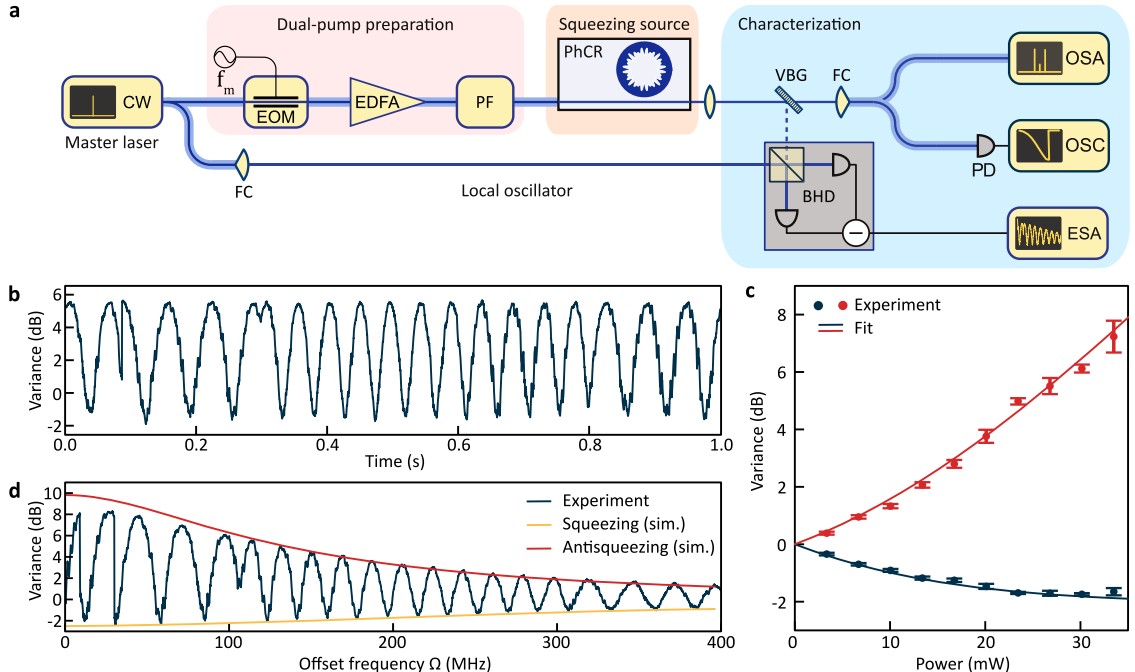

**Fig. 3 | Generation and characterization of single-mode squeezed vacuum (SMSV). a** The dual-pump source is derived from a filtered, electro-optically modulated continuous-wave (CW) master laser and used to pump a photonic crystal ring resonator (PhCR), generating single-mode quadrature squeezing via dual-pump spontaneous four-wave mixing. The generated SMSV is detected by a balanced homodyne detector (BHD) and analyzed using an electronic spectrum analyzer (ESA). EOM electro-optic modulator, EDFA erbium-doped fiber amplifier, PF programmable filter, VBG volume Bragg grating filter, FC fiber collimator, PD photodetector, OSC oscilloscope, OSA optical spectrum analyzer. **b** Measured shot-noise normalized quadrature variance of the SMSV at 27 mW of total pump power and 20 MHz sideband frequency. **c** Measured squeezing and antisqueezing at 20 MHz sideband frequency as a function of total on-chip pump power. Error bars indicate the 1$\sigma$ confidence intervals. **d** Measured squeezing spectrum (corrected for the BHD efficiency and dark noise clearance) at 33 mW of total pump power and simulated squeezing spectrum. All data is taken at 2 MHz resolution bandwidth and 100 Hz video bandwidth.

show, the shape of the squeezing spectrum is well-described by the model presented in the Methods. The parameters of the model are the measured $\kappa$, $\beta$, $\eta$, the pump power, and the inferred total efficiency $T$, with the pump detuning left as a free parameter. The slight discrepancy between the experiment and simulation is attributed to additional nonlinear processes (e.g., cascaded four-wave mixing), asymmetries in the initial pump powers, and resulting differential Kerr shifts.

## Discussion

In conclusion, we demonstrate the generation of single-mode quadrature squeezing based on DP-SFWM in a single PhCR microresonator. The large mode splittings attainable in PhCRs, which are static and immune to disturbances, enable effective suppression of the parasitic nonlinear processes that would otherwise compromise the level of squeezing. We use numerical simulations to find the relevant design parameters and demonstrate experimentally 7.8 dB of estimated on-chip squeezing (in the bus waveguide). We anticipate, this result can be further improved to exceed 10 dB of on-chip squeezing through: (1) enhancing the cavity outcoupling efficiency via pulley couplers; (2) optimizing phase-matching through normal dispersion; (3) improving the cavity $Q$-factor, e.g., by using thinner and wider waveguides cross-section. These improvements are all within the capability of current commercial chip foundry services. The findings presented here highlight the potential of PhCRs for integrated quantum photonics as a scalable, chip-integrated, and CMOS-compatible solution for efficient squeezed light generation. Future work could explore PhCRs for generating quantum states of light in multi-mode systems[43,44,47]. Additionally, integrating PhCR squeezers with on-chip optical parametric amplifiers[48] could mitigate downstream losses, while leveraging synthetic reflection self-injection locking[36] may lead to ultra-compact

power-efficient on-chip sources of low-operational complexity for squeezed light generation. These results open a new route to broad applications of PhCRs in quantum information processing protocols such as Gaussian boson sampling[4,5,49,50], coherent Ising machines[51–53], and cluster state quantum computing[54–56].

## Methods
### Numerical model
To simulate the squeezing spectrum, we consider a system of $N$ driven-dissipative coupled-mode differential equations describing the evolution of the annihilation ($\hat{a}_\mu$) and creation ($\hat{a}_\mu^\dagger$) operators, where $\mu$ denotes the mode number relative to the signal mode $S$ (Fig. 1). We consider a high-$Q$ Kerr-nonlinear microresonator driven by multiple classical, undepleted pumps, each equally detuned from their respective cavity resonances. The corresponding intracavity fields are represented by their complex amplitudes $A_\mu$, while the other cavity modes remain below the oscillation threshold. Under these conditions, the originally quartic interaction Hamiltonian can be linearized and simplified into a quadratic form, so that the evolution of the annihilation and creation operators is given by refs. 41,44:

$$\frac{\partial \hat{a}_\mu}{\partial t} = \sum_\nu R_{\mu\nu}\hat{a}_\nu + \sum_\nu S_{\mu\nu}\hat{a}_\nu^\dagger + \sqrt{\kappa_{ex}}\hat{v}_{ex} + \sqrt{\kappa_0}\hat{v}_0, \quad (1)$$

$$R_{\mu\nu} = -\left[\frac{\kappa}{2} + i(\delta_0 + D_{int}(\nu))\right]\delta(\mu - \nu) + 2ig_0\sum_{j,k}\delta(\mu + j - \nu - k)A_j A_k^*, \quad (2)$$

$$S_{\mu\nu} = ig_0 \sum_{j,k} \delta(\mu + \nu - j - k) A_j A_k. \qquad (3)$$

Where $\delta_0$ is the pump detunings from the respective resonances (equal in our case for both pumps), $g_0$ describes the cubic nonlinearity of the system, $\hat{v}_0$ and $\hat{v}_{ex}$ are the vacuum fluctuations operators.

Equation (1) can now be rewritten in matrix form:

$$\begin{bmatrix} \frac{\partial \mathbf{a}}{\partial t} \\ \frac{\partial \mathbf{a}^\dagger}{\partial t} \end{bmatrix} = \begin{bmatrix} R & S \\ S^* & R^* \end{bmatrix} \begin{bmatrix} \mathbf{a}(t) \\ \mathbf{a}^\dagger(t) \end{bmatrix} + \sqrt{\kappa_0} \begin{bmatrix} \mathbf{v}_0(t) \\ \mathbf{v}_0^\dagger(t) \end{bmatrix} + \sqrt{\kappa_{ex}} \begin{bmatrix} \mathbf{v}_{ex}(t) \\ \mathbf{v}_{ex}^\dagger(t) \end{bmatrix}, \qquad (4)$$

where the vectors of operators $\mathbf{a}$, $\mathbf{v}_0$, and $\mathbf{v}_{ex}$ are constructed from their respective annihilation operators. For example, $\mathbf{a}(t) = (\hat{a}_{-n}(t), \ldots, \hat{a}_n(t))^{\mathsf{T}}$, where $n = (N-1)/2$ for odd $N$ (here: $N = 5$). Using the definitions of position $\hat{x} = (\hat{a} + \hat{a}^\dagger)/\sqrt{2}$ and momentum $\hat{p} = i(\hat{a}^\dagger - \hat{a})/\sqrt{2}$ quadratures, we can now introduce the corresponding time-dependent quadrature vectors:

$$\mathbf{Q} = (\hat{x}_{-n}(t), \ldots, \hat{x}_n(t) | \hat{p}_{-n}(t), \ldots, \hat{p}_n(t))^{\mathsf{T}}, \qquad (5)$$

$$\mathbf{U} = (\hat{x}_{0,-n}(t), \ldots, \hat{x}_{0,n}(t) | \hat{p}_{0,-n}(t), \ldots, \hat{p}_{0,n}(t))^{\mathsf{T}}, \qquad (6)$$

$$\mathbf{V} = (\hat{x}_{ex,-n}(t), \ldots, \hat{x}_{ex,n}(t) | \hat{p}_{ex,-n}(t), \ldots, \hat{p}_{ex,n}(t))^{\mathsf{T}}. \qquad (7)$$

In this case, Eq. (4) is transformed as:

$$\frac{\partial \mathbf{Q}}{\partial t} = \mathbf{M}_q \mathbf{Q} + \sqrt{\kappa_0} \mathbf{U} + \sqrt{\kappa_{ex}} \mathbf{V}, \qquad (8)$$

$$\mathbf{M}_q = \begin{bmatrix} \Re(R + S^*) & -\Im(R + S^*) \\ \Im(R - S^*) & \Re(R - S^*) \end{bmatrix}. \qquad (9)$$

The stationary solution of Eq. (8) can be found in the Fourier domain via matrix inversion. Combined with the cavity input-output relation, this yields:

$$\mathbf{Q}_{out}(\Omega) = \sqrt{\kappa_{ex}} (i\Omega \mathbb{I}_{2N} + \mathbf{M}_q)^{-1} (\sqrt{\kappa_0} \mathbf{U} + \sqrt{\kappa_{ex}} \mathbf{V}) + \mathbf{V}. \qquad (10)$$

Equation (10), along with the commutation relations, can be used to determine the squeezing spectrum.

### Effect of losses

For a specific offset frequency $\Omega$, and assuming an initially pure squeezed state, the variances inside the microresonator of the squeezed and antisqueezed quadratures are given by $V_\pm = e^{\pm 2r}$, where $r$ is the squeezing parameter. Squeezed states, and consequently the level of observed squeezing, are influenced by losses. At the output of an optical channel with efficiency $T$, the quadrature variances are expressed as[6,57]:

$$V_{out, \pm} = T V_\pm + 1 - T.$$

Thus, by measuring both the squeezing and antisqueezing levels, it is possible to infer $r$ and $T$. In the limit of $r \to \infty$ (deterministic bound), the minimal quadrature variance at the cavity output $V_{out,-}$ is equal to $1 - \eta$, where $\eta$ is the cavity outcoupling efficiency.

### Efficiency budget

Independent of inferring $T$ from the measured squeezing values (see previous paragraph), we estimate the total efficiency $T$ based on a

### Table 1 | Efficiency budget for squeezed light detection

| Source of loss | Value |
| --- | --- |
| Ring outcoupling efficiency | 0.9 |
| Chip outcoupling efficiency | 0.94 |
| Mode matching efficiency | 0.81 |
| Optical channel efficiency (after chip) | 0.8 |
| BHD efficiency | 0.75 |
| BHD DNC | 0.99 ($\Omega$ = 0) |

characterization of all sources of loss that the squeezed light experiences before measurement. These include the microresonator out-coupling efficiency, photonic chip outcoupling efficiency (Fresnel loss), volume Bragg grating diffraction efficiency, mode matching efficiency between the local oscillator (LO) and the signal, quantum efficiency of the BHD photodiodes, optical channel loss (after chip), and the BHD's DNC (SI, Section 2). The DNC is equivalent to an optical loss and is frequency-dependent. These factors are summarized in Table 1. Taking all these contributions into account, we estimate that the total efficiency is approximately 0.40 at zero offset frequency $\Omega$.

### Sample fabrication

The samples were fabricated commercially by LIGENTEC SA using UV optical lithography. The coupling waveguide has the same width as the ring waveguide, and the coupling gap is 300 nm.

### Data availability

The data shown in the plots are available through the Zenodo repository: https://doi.org/10.5281/zenodo.17349057.

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

## Acknowledgements

This project has received funding from the European Research Council (ERC, grant agreement No 853564, T.H.), from the European Innovation Council (EIC, grant agreement No 101137000, T.H.) and through the Helmholtz Young Investigators Group (VH-NG-1404, T.H.); the work was supported through the Maxwell computational resources operated at DESY.

## Author contributions

A.U. designed the setup and the photonic chip, performed the experiments, and analyzed the data. T.W. and B.R. supported the design of the

resonator and the experiments. T.H. supervised the work. A.U. and T.H. prepared the manuscript with input from all authors.

## Funding

## Competing interests

The authors declare no competing interests.
