## [Transparent Peer Review file · Nature Communications]

Quadrature squeezing in a nanophotonic microresonator

Corresponding Author: Professor Tobias Herr

Version 0:

Reviewer comments:

Reviewer #1

(Remarks to the Author)

The manuscript presents a $\chi^{(3)}$ squeezed light source based on degenerate dual-pump spontaneous four-wave mixing (DP-SFWM) in a silicon nitride (Si_3N_4) platform. A key innovation is the use of a photonic crystal ring resonator (PhCR) to engineer mode splitting at the $m_0 \pm 2$ resonances, which serves to suppress parasitic nonlinear processes that would otherwise introduce noise and limit squeezing efficiency. The dual-pump configuration is implemented via an electro-optic modulator (EOM) driven at the frequency separation between the pump modes; the same laser is split to serve as the local oscillator (LO) for balanced homodyne detection (BHD).

While the application of PhCR for suppression of competing nonlinearities is elegant and technically sound, the manuscript does not appear to present a fundamentally new mechanism or a technically significant advance. Below are several specific concerns:

1. Results are presented from a single PhCR device, with no direct comparison to conventional microring resonators or alternative configurations. Including such comparisons is essential to establish the unique benefit of the PhCR design.
2. The reported 7.8 dB of on-chip squeezing is inferred rather than directly measured. The inference is based on fitting a model using a single pair of squeezing and anti-squeezing values ($V_+ = 5.54$ dB, $V_- = 1.71$ dB) to extract the transmission coefficient $T = 0.37$ and the squeezing parameter r . Further justification for the selected parameters, along with an analysis of how T and r vary with pump power and analysis frequency, is needed. Confidence intervals or error bounds on the inferred squeezing would help substantiate the claim.
3. Only intrinsic and coupling losses are considered, while parasitic losses—potentially significant limiting factors—are neglected. A quantitative discussion or estimate of parasitic loss would improve the analysis.
4. The manuscript does not characterize the measured vacuum noise. It is unclear whether the observed squeezing is truly vacuum squeezing or includes suppression of technical noise or laser thermal noise. Clarification on pump noise assessment and justification for attributing the squeezing to vacuum fluctuations is needed.
5. The suppression mechanism is based on a mode splitting of $\beta \approx 13.5$ (5.4 GHz). The authors should elaborate on: (i) whether increasing the splitting could extend the squeezing bandwidth, and the associated trade-offs (e.g., coupling efficiency, dispersion, or finesse); and (ii) how the PhCR approach compares in suppression efficiency to prior techniques such as coupled resonators.
6. The optical parametric oscillation (OPO) threshold is not reported. Including this information would provide valuable context for understanding the device's operating regime.
7. The method for generating a dual-tone pump with ~ 300 GHz separation using an EOM is unclear and requires additional explanation.

In summary, while the design is thoughtful and the experimental work is technically strong, the manuscript does not present results that are sufficiently novel or compelling to warrant publication in Nature Communications in its current form.

[1] Dutt, A. et al., Science Advances, 6(25), eaba9186 (2020). <https://www.science.org/doi/10.1126/sciadv.aba9186>

Reviewer #2

(Remarks to the Author)

This work from Herr' group presents the generation of degenerate quadrature squeezing in an integrated photonic chip using a nano-corrugated photonic crystal ring (PhCR) microresonator. By employing tailored dispersion engineering to suppress parasitic nonlinear processes, the authors enhance the efficiency of dual-pump four-wave mixing, thereby optimizing squeezed-light generation. Through dual-pump spontaneous four-wave mixing (DP-SFWM), they achieve an estimated 7.8

dB of on-chip squeezing in a CMOS-compatible silicon nitride platform. A key innovation is the PhCR's static corrugation pattern, which modifies the resonator's mode structure to suppress parasitic nonlinear processes—single-pump spontaneous four-wave mixing (SP-SFWM) and Bragg-scattering four-wave mixing (BS-FWM)—without requiring active thermal or electrical tuning. This design simplifies operation in uncontrolled environments and aligns with scalable foundry fabrication processes. The authors demonstrate broadband squeezing and propose pathways to further improve squeezing levels (>10 dB) through enhanced outcoupling efficiency and dispersion engineering. This work highlights nanophotonic PhCRs as promising candidates for applications such as quantum-enhanced sensing, quantum boson sampling, and optical quantum computing.

The study reports a novel and interesting approach to generating vacuum squeezed light on integrated photonic chips, and the experimental results are of high technical quality. However, I have some concerns regarding the breadth of scientific and technological novelty in this work.

First, as the authors acknowledge, similar PhCR structures have been used for non-quantum applications in Refs. 35–40, and the same group has employed this device for microcomb generation in Refs. 36 and 37. Could the authors clarify the specific novelty and technological advancements that enable light squeezing in this context?

While the results are commendable, the manuscript could better contextualize its advances relative to existing strategies for parasitic noise suppression in integrated squeezers. Notably, Ref. 23 (Nature Communications 12, 2233, 2021) achieves comparable on-chip squeezing (8 dB) using a dynamically tunable photonic molecule architecture. The two-resonator design in Ref. 23 hybridizes and detunes parasitic modes via auxiliary ring coupling, enabling optimized noise suppression.

In contrast, the static nano-corrugation design in this work introduces a trade-off between structural simplicity and adaptable performance. While the single-resonator architecture avoids challenges inherent to multi-resonator systems—such as thermal crosstalk between coupled rings (Nature Communications 12, 2233, 2021)—the fixed corrugation pattern rigidly locks resonance splitting characteristics. This rigidity may limit the suppression efficiency of parasitic nonlinear processes to a predetermined range, potentially constraining the maximum achievable squeezing levels compared to dynamically tunable systems like those in Refs. 23 and 33. For instance, Ref. 23's photonic molecule achieves 8 dB squeezing by actively tuning coupled resonators, whereas this work's static mechanism, while achieving 7.8 dB, may not fully optimize nonlinear interactions under fabrication variations or environmental fluctuations.

The lack of tunability becomes particularly critical for scalable applications such as continuous-variable quantum computing, where large-scale integration demands uniformity across multiple squeezed light sources. Chip-scale fabrication variations—inevitable in wafer-level processes—could lead to performance discrepancies between nominally identical squeezers. In dynamically tunable systems (Refs. 23 and 33), thermal or electro-optic tuning can compensate for such variations, ensuring uniform squeezing levels and spectral alignment. In contrast, this work's static design lacks adaptability: once fabricated, its resonance splitting is immutable, risking mismatched squeezing parameters between sources. For quantum computing applications requiring synchronized multi-mode squeezing or cluster-state generation, even minor deviations in squeezing levels or spectral overlap could degrade entanglement fidelity and computational accuracy. While the simplicity of this design aligns with CMOS compatibility, its fixed operational regime may hinder scalability in error-sensitive quantum applications. I would appreciate it if the authors could clarify how they plan to address this scalability issue.

Additionally, several experiments demonstrated the generation of multimode squeezing/Gaussian states on the same silicon nitride platform (Nature 639, 329–336, 2025; Nature 638, 912–919, 2025). Could the authors comment on and discuss how their PhCR squeezers could be scaled up for multimode operation?

I hope the authors can address these literature-related concerns and clarify the technical advancements and scalability issues before I finalize my recommendation regarding publication.

Reviewer #3

(Remarks to the Author)

Version 1:

Reviewer comments:

Reviewer #1

(Remarks to the Author)

I appreciate that the authors have addressed the technical improvement questions I raised in my previous review. However, I remain unconvinced that this work constitutes a significant advance. In my view, the results primarily demonstrate structural variations relative to the current state of the art, rather than a substantive breakthrough.

Inferred squeezing does not amount to a direct assessment of actual squeezing, as the conclusions are confounded by multiple interdependencies. The lack of tunability ultimately places the work at a disadvantage compared to existing

approaches, such as photonic molecules.

Reviewer #2

(Remarks to the Author)

The authors have done a good job addressing the reviewers' comments. The revisions have significantly strengthened the manuscript. The expanded functionalities demonstrated here surpass prior work, but also highlight the novelty and impact of this study. I am pleased to recommend this paper for publication in its present form.

Reply to the Reviewers

We thank all Reviewers for their careful assessment of our work, their insightful comments, and constructive criticism. The Reviewers' comments have led to a substantially revised manuscript with improved clarity and additional data. We hope the revisions, along with our detailed reply below, satisfactorily address all concerns raised by the Reviewers.

- In this reply letter, the descriptions of revisions are underlined.
- In the manuscript, all revisions are highlighted in blue color.
- SI refers to the new Supplementary Information document.

Reviewer #1 (Remarks to the Author):

The manuscript presents a $\chi^{(3)}$ squeezed light source based on degenerate dual-pump spontaneous four-wave mixing (DP-SFWM) in a silicon nitride (Si_3N_4) platform. A key innovation is the use of a photonic crystal ring resonator (PhCR) to engineer mode splitting at the $m_0 \pm 2$ resonances, which serves to suppress parasitic nonlinear processes that would otherwise introduce noise and limit squeezing efficiency. The dual-pump configuration is implemented via an electro-optic modulator (EOM) driven at the frequency separation between the pump modes; the same laser is split to serve as the local oscillator (LO) for balanced homodyne detection (BHD).

While the application of PhCR for suppression of competing nonlinearities is elegant and technically sound, the manuscript does not appear to present a fundamentally new mechanism or a technically significant advance. Below are several specific concerns:

We thank the Reviewer for the careful and constructive report, and acknowledging the PhCR approach as elegant and technically sound. We also understand that the Reviewer would like the manuscript to better explain why our approach represents a significant advance. To this end, we will address each point below in detail and describe the changes we made in the manuscript.

1. Results are presented from a single PhCR device, with no direct comparison to conventional microring resonators or alternative configurations. Including such comparisons is essential to establish the unique benefit of the PhCR design.

We thank the Reviewer for this suggestion and agree that a comparison with conventional microrings and alternative configurations is useful. In the table below, we compare our result to previous studies on CMOS-compatible single-mode quadrature squeezing:

Ref	Pump power, mW	Squeezing in the cavity, dB
Zhao et al. PRL 2020	102	6.26
Zhang et al. Nat. Comm. 2021	83	9.3
This work	33	11.3

Thus, in absolute numbers, our approach outperforms previous demonstrations. However, as many concept-independent factors – such as propagation loss, coupling conditions, or

fabrication quality – determine the performance of the devices, we believe the numbers given above would not provide a meaningful comparison of the underlying approaches. Instead, we would like to provide a more conceptual comparison of the different approaches:

- (i) Symmetric pump detuning can mitigate parasitic nonlinear processes (Zhao et al. PRL 2020), but this approach comes at the cost of reduced pump efficiency.
- (ii) Coupled resonators (Zhang et al. Nat. Commun. 2021), where mode hybridization detunes parasitic modes in frequency are more power efficient, and effective at suppressing parasitic nonlinear processes. However, it requires active thermal tuning and stabilization of the coupled resonators. A precise FSR ratio between the primary and auxiliary resonators is also crucial and imposes strict fabrication tolerances.
- (iii) PhCRs, as demonstrated in this work, can achieve strong mode hybridization through static, lithographically defined corrugation patterns. These introduce intrinsically stable and rigid resonance frequency shifts that suppress parasitic nonlinear processes without the need for active control. Thus, PhCRs are immune to environmental or pump-induced fluctuations variation, which is a unique feature and favorable for easily operable, passively stable, and scalable systems.

In summary, while all approaches can achieve parasitic mode suppression, the PhCR platform is both power efficient and inherently stable. To clarify this point, we have revised the manuscript (lines 68, 88, and 290) to better differentiate the 3 approaches.

2. The reported 7.8 dB of on-chip squeezing is inferred rather than directly measured. The inference is based on fitting a model using a single pair of squeezing and anti-squeezing values ($V_+ = 5.54$ dB, $V_- = 1.71$ dB) to extract the transmission coefficient $T = 0.37$ and the squeezing parameter r . Further justification for the selected parameters, along with an analysis of how T and r vary with pump power and analysis frequency, is needed. Confidence intervals or error bounds on the inferred squeezing would help substantiate the claim.

We appreciate the Reviewer's request to clarify the analysis including (i) the frequency dependence of T and r , (ii) the power dependence of T and r , and (iii) error bounds on the inferred squeezing. We will discuss all aspects below:

Both T and r are dependent on the analysis or offset frequency Ω . Although the material losses, resonator coupling, Fresnel losses etc. can be considered essentially independent of frequency across the optical interval that corresponds to the range of the analysis frequency, the BHD is frequency dependent, which can be characterized via the BHD dark noise clearance (DNC). The frequency dependence of r originates largely from the resonance shape, but may also be impacted by a potential pump detuning or cavity dispersion.

Thus, the relations $V_{out,\pm} = TV_{\pm} + 1 - T$ and $V_{\pm} = e^{\mp 2r}$, which we introduced in the Methods and which we have used in our analysis hold separately for each

specific offset frequency Ω . Based on the assumption of an initially pure squeezed vacuum state that passes through an optical channel with power transmission T , they enable connecting the measures $V_{out,\pm}$ to T and r , and thereby to the intracavity squeezing V_{\pm} (refs. 6, 58).

To clarify this point, we have revised the manuscript (lines 151 and 353) to clearly point out the frequency dependence of the squeezing level and that the relations quoted above holds separately for each specific offset frequency. Moreover, we clarify in the caption of Figure 3, as well as in the main text (line 266) that the data show in Figure 3d has been corrected for the frequency dependent BHD detection (BHD quantum efficiency and DNC) to reproduce the shape of the actual squeezing spectrum prior to detection (this practically removes the frequency dependence of T , so that the frequency dependence of r is apparent). In the new SI, Sec. 2 we provide more detail on the frequency dependence of the DNC.

- (i) In our manuscript we had already shown the power dependence of the achieved squeezing close to DC ($\Omega = 20$ MHz) in Fig. 3c. However, we welcome the suggestion by the Reviewer to complement this data, with the power dependence of the efficiency T and the squeezing parameter r (see Fig. 1 in this reply document). We observe that T decreases slightly with increasing pump power, which we attribute to slightly increased influence of cascaded four-wave mixing as well as parasitic nonlinear processes such as SP-SFWM and BS-FWM at higher pump power. Further details on estimating the impact of parasitic processes are discussed in the response to the next comment. While the efficiency T does not show a significant power dependence, the squeezing parameter r (at $\Omega = 20$ MHz) increases linearly with pump power, as expected from the Kerr Hamiltonian term responsible for the squeezing process.

To address this point, we have revised the manuscript (line 254) and added Fig. 1 of this document as a new figure in the SI, along with a description (SI, Sec. 4).

Figure 1. Total efficiency T and squeezing parameter r in dependence on the total pump power, measured at an offset frequency of $\Omega = 20$ MHz.

- (ii) We agree with the Reviewer, that error bounds on the inferred squeezing values are important. The uncertainty in the cavity linewidth measurement is approximately 3.7% (impacting the ring outcoupling efficiency), as derived from linewidth fitting. The statistical uncertainty of measuring the squeezing and anti-squeezing levels is approximately 5.8%. Propagating these dominating uncertainties, based on the analytic expression for on-chip squeezing in the Methods, this results in an uncertainty of ± 1.1 dB in our reported estimate of 7.8 dB for the squeezing in the bus waveguide.

We have revised the manuscript accordingly, added uncertainties to the reported squeezing levels (line 270), and included error-bars in Fig. 3c.

3. Only intrinsic and coupling losses are considered, while parasitic losses—potentially significant limiting factors—are neglected. A quantitative discussion or estimate of parasitic loss would improve the analysis.

We thank the Reviewer for this insightful remark and agree that an estimate of the impact of parasitic processes would strengthen the manuscript.

By considering the (linear optical) loss budget as presented in the Methods – including outcoupling, transmission, and detection losses, we calculated a total detection efficiency of approximately 40 % with an estimated uncertainty of 3 %, dominated by the power meter reading. From the measured squeezing and anti-squeezing values, we infer a combined outcoupling, collection, and detection efficiency of ~ 37 %. The latter efficiency value represents a lower-bound estimate, under the assumption that the initially generated squeezed state is pure (i.e., no additional parasitic processes). Given that both values are consistent, we conclude that the impact of additional parasitic processes is minor.

To corroborate the minor impact of the parasitic processes in our system, we conducted additional numerical simulations based on the model presented in the Methods. The simulation enables us to compare the scenarios with and without parasitic processes. These simulations, which we further detail in the new SI Sec. 3, confirm that the impact of parasitic processes is indeed small and equivalent to an additional effective loss of approximately 1.6 %. This is shown in Fig. 2 in this document: the horizontal black dashed line shows the case without parasitic processes and assuming $T = 90\%$ efficiency. The dots correspond to a numeric simulation including parasitic processes (based on the framework presented in Methods). For the specific value of $\beta \approx 13.5$ in our experiment, we see a slightly reduced squeezing level compared to the black dashed line. Changing the efficiency to $T = 88.6\%$ and assuming absence of parasitic processes, we obtain the red dashed line, which intersects with our experimental system. Thus, we can estimate that the effective relative loss resulting from unsuppressed parasitic nonlinear processes is small and below $1 - 0.886/0.9 \approx 1.6\%$.

Figure 2. Numerically simulated squeezing at zero frequency offset (blue dots) for different normalized mode splitting β and theoretical squeezing levels corresponding to quantum efficiencies of 0.9 and 0.886 (black and red dashed lines respectively).

To address this point, we have added Figure 2 of this document to the new SI Sec. 3, where we present this estimate on the impact of parasitic processes. The main text has been revised accordingly (line 172, 207, and 254).

4. The manuscript does not characterize the measured vacuum noise. It is unclear whether the observed squeezing is truly vacuum squeezing or includes suppression of technical noise or laser thermal noise. Clarification on pump noise assessment and justification for attributing the squeezing to vacuum fluctuations is needed.

We thank the Reviewer for this important comment. Indeed, we had performed a full characterization of the vacuum noise level and dark noise clearance (DNC) of the BHD.

Motivated by the Reviewer's comment we have including these measurements to the manuscript.

Figure 3. Balanced homodyne detector electronic, shot noise spectra, and dark noise clearance (DNC).

We follow the standard procedure in the field to confirm that the measured squeezing originates from vacuum fluctuations and is not influenced by technical noise from the pump laser. Specifically, we performed the following steps:

- (i) With the local oscillator (LO) blocked, we recorded the electronic noise floor of the BHD (Fig. 3 in this reply document, blue line). This establishes the baseline noise level of the detection electronics.
- (ii) With the LO open and no signal (vacuum) at the input port of the BHD, we measured the shot noise level (Fig. 3 in this reply document, red line). This measurement defines the vacuum noise reference for subsequent squeezing measurements.
- (iii) By comparing the electronic noise and shot noise spectra, we quantified the DNC of the detector. In our setup, the DNC is above 20 dB at low frequency offsets and ~ 10 dB at high offsets. These values ensure that the observed squeezing is not an artifact of detector noise.
- (iv) The LO and pump beams are derived from the same stable continuous-wave laser, and the homodyne detection is performed in a mode-matched, balanced configuration. This configuration inherently rejects common-mode technical noise from the laser.

To address this point, we have added the measurements present in Fig. 3 of this document to the new SI, Sec. 2 along with a description on how the data was obtained. A reference to the SI was made in the main text (line 376).

5. The suppression mechanism is based on a mode splitting of $\beta \approx 13.5$ (5.4 GHz). The authors should elaborate on: (i) whether increasing the splitting could extend the squeezing bandwidth, and the associated trade-offs (e.g., coupling efficiency, dispersion, or finesse); and (ii) how the PhCR approach compares in suppression efficiency to prior techniques such as coupled resonators.

We thank the Reviewer for these valuable questions and will reply to both points (i) and (ii):

- (i) The amount of splitting defines how well parasitic processes are suppressed, but it is not suitable to increase the squeezing bandwidth. The reason is that the split modes are not part of the (single mode) squeezing process. Indeed, the squeezing bandwidth is in our case largely determined by the cavity linewidth. While this is not the focus of the present work, we note that we have validated in ref. 37 for a similar resonator that the intrinsic linewidth (and with it, finesse and coupling efficiency) remain virtually unchanged, up to mode splitting of several tens of GHz.

To address this point we highlight in the revised manuscript (line 207) the opportunity for strong splitting (Fig. 2d), without significant loss penalty ref. 37.

- (ii) In the reply to the first point, we already contrasted our approach to prior techniques. We appreciate the opportunity to provide here a more detailed discussion: Both coupled-resonator systems and PhCRs suppress parasitic nonlinear interactions by detuning the resonance frequencies of modes involved in unwanted processes. In both cases, for a given resonator linewidth, their suppression efficiency ultimately depends on the achievable magnitude of the mode frequency shifts/splitting. However, the coupled-resonator systems and PhCRs approaches differ in the underlying mechanisms used to realize these mode shifts: In coupled-resonator systems (photonic molecules), suppression relies on mode hybridization between approximately degenerate modes in two distinct ring resonators. To achieve the mode structure with the two suitably split modes, this approach requires careful engineering and fabrication of the FSR ratio, and efforts to maintain a suitable detuning between the involved resonances. As a result, coupled resonators are highly sensitive to differential thermal shifts and fabrication imperfections – such as variations in layer thickness – that can alter group velocity dispersion and induce deviation from the intended FSR ratio, impacting the suppression of parasitic processes. In contrast, PhCRs achieve resonance splitting via controlled coupling between the intrinsically frequency-degenerate clockwise and counter-clockwise modes. Their mutual coupling is engineered through periodic corrugations in the resonator, with the split modes number and splitting strength statically inscribed by the corrugation pattern. The PhCR approach is inherently robust to fabrication imperfections. While layer thickness variations may shift the absolute resonance frequencies, the angular mode numbers targeted by the corrugation pattern remain unchanged. Therefore, the corrugation pattern will continue to split the correct resonances with the same amount of splitting, even in the presence of environmental or operational fluctuations. The key working principle in a PhCR is thus immune to disturbances. Fabrication-induced variations in corrugation pattern contrast may

reduce the splitting amplitude slightly. However, if the splitting is sufficiently large, small deviations from the nominal value will not impact noticeably the system's performance, as the suppression of parasitic processes (naturally) saturates (see Fig. 2 of this document).

We believe the combined revisions made in response to points 1, 2, and 3 in both main text and SI already address this point.

6. The optical parametric oscillation (OPO) threshold is not reported. Including this information would provide valuable context for understanding the device's operating regime.

We thank the Reviewer for highlighting this important aspect. To address it, we performed additional measurements to accurately determine the degenerate OPO threshold. Specifically, we measured the generated OPO power as a function of the pump power. These measurements are done using a high-sensitivity optical spectrum analyzer operating in zero-span mode, centered at the signal wavelength. The resulting data (Fig. 4 in this document) follow the exponential dependence predicted by the dual-pump DOPO theory (ref. 19). From this data, we extract an DOPO threshold of 39.36 ± 2.19 mW. The uncertainty results from the combined uncertainties of both the measured pump power and the DOPO power.

Figure 4. Degenerate optical parametric oscillator threshold power characterization with exponential fit.

To address this point, we have included this experimental data in the new SI, Sec. 5 and revised the Main text (line 261).

7. The method for generating a dual-tone pump with ~ 300 GHz separation using an EOM is unclear and requires additional explanation.

We thank the Reviewer for pointing out this omission. Indeed, the original manuscript did not sufficiently explain our method for generating the dual-pump light. To generate the dual-pump with approximately 600 GHz line separation, we use an electro-optic comb (EOC) generator consisting of two cascaded electro-optic phase modulators. A tunable continuous-wave laser is sent through this setup, producing an optical frequency comb with line spacing of ~ 16.67 GHz, spanning approximately 733 GHz (44 sidebands). The EOC is sent to a programmable optical filter, which is configured to transmit only ~ 20 GHz spectral windows centered on the ± 18 th sidebands of the EOC. Effectively, this setup generates two optical sidebands centered symmetrically around the original master laser frequency. Those two sidebands are amplified using an EDFA. Crucially, amplified spontaneous emission at the master laser wavelength is suppressed using a fiber-based Bragg grating, before coupling the dual-pump into the photonic chip.

To address this comment, we have included detailed information on the EOC-based dual pump source in the SI, Sec. 1 and referenced it in the Main text (line 218).

In summary, while the design is thoughtful and the experimental work is technically strong, the manuscript does not present results that are sufficiently novel or compelling to warrant publication in Nature Communications in its current form.

[1] Dutt, A. et al., Science Advances, 6(25), eaba9186 (2020). <https://www.science.org/doi/10.1126/sciadv.aba9186>

We thank the Reviewer for pointing out the reference. It was already included in the first version of the manuscript. The name of the first author of the paper is Vaidya, not Dutt. Perhaps this has led to confusion.

In sum, we are grateful to the Reviewer for the constructive and helpful comments, which have led to major revision, new analysis, additional experiments, and a new SI. We believe the revision we made in response to the Reviewer's comments have strengthened and completed our manuscript. Considering our revisions, we hope that the Reviewer can recommend our revised manuscript for publication.

Reviewer #2 (Remarks to the Author):

This work from Herr' group presents the generation of degenerate quadrature squeezing in an integrated photonic chip using a nano-corrugated photonic crystal ring (PhCR) microresonator. By employing tailored dispersion engineering to suppress parasitic nonlinear processes, the authors enhance the efficiency of dual-pump four-wave mixing, thereby optimizing squeezed-light generation. Through dual-pump spontaneous four-wave mixing (DP-SFWM), they achieve an estimated 7.8 dB of on-chip squeezing in a CMOS-compatible silicon nitride platform. A key innovation is the PhCR's static corrugation pattern, which modifies the resonator's mode structure to suppress parasitic nonlinear processes—single-pump

spontaneous four-wave mixing (SP-SFWM) and Bragg-scattering four-wave mixing (BS-FWM)—without requiring active thermal or electrical tuning. This design simplifies operation in uncontrolled environments and aligns with scalable foundry fabrication processes. The authors demonstrate broadband squeezing and propose pathways to further improve squeezing levels (>10 dB) through enhanced outcoupling efficiency and dispersion engineering. This work highlights nanophotonic PhCRs as promising candidates for applications such as quantum-enhanced sensing, quantum boson sampling, and optical quantum computing.

The study reports a novel and interesting approach to generating vacuum squeezed light on integrated photonic chips, and the experimental results are of high technical quality. However, I have some concerns regarding the breadth of scientific and technological novelty in this work.

We are happy to learn that the Reviewer finds our work novel and of high technical quality, and we thank the Reviewer for the careful and constructive report. We also acknowledge and appreciate the concerns raised by the Reviewer which we address point-by-point below.

First, as the authors acknowledge, similar PhCR structures have been used for non-quantum applications in Refs. 35–40, and the same group has employed this device for microcomb generation in Refs. 36 and 37. Could the authors clarify the specific novelty and technological advancements that enable light squeezing in this context?

We thank the Reviewer for this comment and request for clarification. PhCR resonators are a novel yet highly versatile class of microresonator that is attracting increasing interest, especially in nonlinear photonics. In addition to dispersion engineering through waveguide cross-section tailoring, they enable the controlled hybridization of forward and backward propagating modes. In soliton microcombs, this has made it possible to shift the parametric threshold into the soliton attractor regime for direct pulse formation (ref. 35), single pulse formation, and deterministic backward wave generation for self-injection locking and stabilization (refs 36 and 37). PhCRs have also enabled localized and slow light by impacting group velocity (ref. 38), phase matching for optical parametric oscillation with large spectral separation (ref. 39), and shaping of the spectral envelope in frequency combs (ref. 40). These are all classical applications where efficient frequency conversion into coherent, above threshold oscillation is the goal. In the present work, the opposite is the case. Indeed, we utilize the PhCR to suppress coherent oscillation and unwanted nonlinear processes that would otherwise obstruct the sub-threshold squeezed state of light.

To achieve this, our system implements large mode frequency shifts through strong coupling between counterpropagating modes (coupling rate more than two orders of magnitude higher than the intrinsic cavity decay rate). These mode shifts detune the modes with index $m = \pm 2$, and prevent them from supporting unwanted nonlinear processes. Critically, these strong frequency shifts are simultaneously implemented for two modes in the (spectral) vicinity to the mode in which the squeezed light is generated, and where the absence of any parasitic mode splitting is crucial (otherwise squeezing would be generated in a hybridized mode basis, which cannot be coupled out efficiently). Achieving such high-contrast splitting (large splitting and no splitting) in nearby modes, on a level that is compatible with the demonstrated quantum application is not at all obvious in a PhCR and has not been demonstrated in prior work.

To address this point, we have revised the manuscript (line 207), highlighting the large dual splitting is attainable in PhCRs in a controlled manner and while maintaining low loss.

While the results are commendable, the manuscript could better contextualize its advances relative to existing strategies for parasitic noise suppression in integrated squeezers. Notably, Ref. 23 (Nature Communications 12, 2233, 2021) achieves comparable on-chip squeezing (8 dB) using a dynamically tunable photonic molecule architecture. The two-resonator design in Ref. 23 hybridizes and detunes parasitic modes via auxiliary ring coupling, enabling optimized noise suppression. In contrast, the static nano-corrugation design in this work introduces a trade-off between structural simplicity and adaptable performance. While the single-resonator architecture avoids challenges inherent to multi-resonator systems—such as thermal crosstalk between coupled rings (Nature Communications 12, 2233, 2021)—the fixed corrugation pattern rigidly locks resonance splitting characteristics. This rigidity may limit the suppression efficiency of parasitic nonlinear processes to a predetermined range, potentially constraining the maximum achievable squeezing levels compared to dynamically tunable systems like those in Refs. 23 and 33. For instance, Ref. 23's photonic molecule achieves 8 dB squeezing by actively tuning coupled resonators, whereas this work's static mechanism, while achieving 7.8 dB, may not fully optimize nonlinear interactions under fabrication variations or environmental fluctuations.

We agree with the Reviewer that coupled ring resonators ('photonic molecules') are a powerful platform for controlling nonlinear interactions. They have several advantages, particularly for comb generation, as it enables transitioning between the non-hybridized and hybridized mode regimes. For example, this enabled combining low-threshold system start-up with high-efficiency coherent oscillation; see Helgason et al., Nature Photonics, 2023, <https://doi.org/10.1038/s41566-023-01280-3>).

As the Reviewer points out, a challenge in multi-resonator systems is thermal crosstalk between coupled rings. Even more problematic is the fragile operating state of photonic molecules, which requires careful active alignment and stabilization. For instance, considering a typical thermorefractive coefficient of 10^{-5} K^{-1} and a resonance width of 100 MHz, a few milli-Kelvin temperature stability is needed to keep differential resonance drifts below a tenth of the linewidth. This must be achieved and sustained while coupling strong pump lasers to only one of the resonators (ref. 23, Nature Communications 12, 2233, 2021). Active stabilization is certainly manageable for one or a few resonators, but becomes increasingly challenging for a larger number of resonators, as required for large-scale computing applications. Furthermore, to achieve strong mode hybridization in the two parasitic modes (while minimally affecting the signal mode), the two resonators in the photonic molecule, must precisely maintain a very specific FSR ratio. This constraint imposes strict fabrication tolerances, particularly in terms of layer thickness and dispersion engineering.

In contrast, and as the Reviewer also points out, the mode hybridization in PhCRs lacks differential tunability, implying that the mode frequency shifts are rigid and fixed by design. While this may reduce flexibility in some scenarios, it is ideal for squeezed light generation, as reported here; it maintains an optimal working point in a simple, easily operable architecture.

Importantly, no precise value of the hybridization strength needs to be achieved, as long as the induced mode frequency shifts are strong enough to suppress the unwanted processes.

To clarify this, we have added a discussion and a figure to the SI (Fig. 2 in this document) that illustrates how a sufficiently high splitting insures a high squeezing. In particular, the figure shows that the squeezing level converges to the deterministic bound, i.e. for sufficiently high splitting, the squeezing changes only weakly with β . Thus, unlike a photonic molecule, which requires precise fabrication and tuning to achieve strong hybridization, a PhCR is tolerant to fabrication inconsistencies and fluctuations in the operation conditions.

The lack of tunability becomes particularly critical for scalable applications such as continuous-variable quantum computing, where large-scale integration demands uniformity across multiple squeezed light sources. Chip-scale fabrication variations—inevitable in wafer-level processes—could lead to performance discrepancies between nominally identical squeezers. In dynamically tunable systems (Refs. 23 and 33), thermal or electro-optic tuning can compensate for such variations, ensuring uniform squeezing levels and spectral alignment. In contrast, this work's static design lacks adaptability: once fabricated, its resonance splitting is immutable, risking mismatched squeezing parameters between sources. For quantum computing applications requiring synchronized multi-mode squeezing or cluster-state generation, even minor deviations in squeezing levels or spectral overlap could degrade entanglement fidelity and computational accuracy. While the simplicity of this design aligns with CMOS compatibility, its fixed operational regime may hinder scalability in error-sensitive quantum applications. I would appreciate it if the authors could clarify how they plan to address this scalability issue.

We appreciate this comment by the Reviewer, touching indeed upon a critical point for scalability. Indeed, fabrication across multiple chips or even on one chip will always yield slight fabrication deviations, preventing identical resonators. Here we see indeed a strong advantage of the PhCR approach when compared to photonic molecules. The aspects of operational stability and the requirement of perfectly matched and aligned rings for mode-hybridization in photonic molecules have been discussed above and are not repeated here. Here, we discuss first two related points (i) and (ii), before addressing the core of the Reviewer's question under (iii):

- (i) *Wavelength of the squeezed light:* The wavelength of the squeezed light is defined by the resonance frequency of the mode in which squeezing occurs. It can be adjusted via the temperature of the squeezing resonator. This applies to both photonic molecules and PhCRs in the same way.
- (ii) *Alignment of the pumped resonances with the two pump lasers:* This may become relevant if many squeezers are to be pumped by a single pair of pump lasers. Assuming that the signal modes are wavelength aligned (see (i)), the pump mode alignment cannot differ by more than the difference of the free-spectral range (FSR). As this depends on the fabrication process it is challenging to make a general statement here, but again, this applies to both photonic molecules and PhCRs in the same way. Based on our experience with a commercial foundry process, the FSR differences between resonators

are usually small compared to the linewidth of the cavity. Thus, after alignment of the signal modes (see (i)), it should be possible to pump a larger number of squeezers with the same pair of pump lasers. We note that any small deviation here is much less critical in comparison to the precise need for alignment of the FSR in the photonic molecule-based system. Slightly off-resonant pumping from imperfect FSR values, can however reduce the effective pump power and squeezing level (see point (iii)).

- (iii) *Adjusting the level of squeezing:* This point has been highlighted by the Reviewer and indeed nominally identical squeezers, even if the squeezing modes are perfectly frequency aligned (see (i)), may still differ in the squeezing level, for instance through a small difference in FSR and therefore imperfect pump mode alignment (see (ii)). However, we would like to highlight, that a precise value of the mode splitting is uncritical, as long as the induced mode frequency shifts are strong enough to suppress the unwanted processes (see reply to the previous comment). This makes PhCRs a robust, fabrication imperfection tolerant and hence scalable platform for squeezing. Should it indeed become necessary to adjust and equalize the level of squeezing, this can be done by adjusting the pump power that is coupled to each of the squeezers. For instance, this can be accomplished with thermally or electro-optically tuned Mach-Zehnder interferometers, which are robust against temperature fluctuation (owing to the wide fringe width compared to a resonance width), and which are routinely used in chip-integrated quantum photonics.

To address this point, we have clarified explicitly that level of squeezing depends on the pump power in line 254 of the revised manuscript. We also clarified the thermal tuning of the resonator can be pursued, rather than tuning the pump lasers (line 235).

Additionally, several experiments demonstrated the generation of multimode squeezing/Gaussian states on the same silicon nitride platform (Nature 639, 329–336, 2025; Nature 638, 912–919, 2025). Could the authors comment on and discuss how their PhCR squeezers could be scaled up for multimode operation?

We thank the Reviewer for highlighting these references; one of them was indeed missing and we have added it.

The PhCRs could be readily utilized in both types of multimode squeezed light experiments. For instance, with regards to Aghaee Rad et al. (Nature 638, 912–919, 2025) the inherent stability of the PhCRs could be beneficial, permitting even larger scale up. As discussed earlier, this could significantly simplify integration, reduce thermal crosstalk, and enhances the overall operational stability of the system without compromising achievable squeezing levels.

In the context of frequency-degrees of freedom (Jia et al. , Nature 639, 329–336, 2025), PhCRs also offer unique opportunities. Crucially, multiple spatial frequency (Fourier) components of the corrugation can be superimposed within a single PhCR (see ref. 40). Each of these components selectively targets a specific mode number, allowing for

parallel mode hybridization with individually programmable strength. This capability can be exploited to engineer mode-selective interactions in the frequency domain and enable or suppress particular nonlinear processes. Such control, challenging to achieve by coupling distinct resonators together, is highly desirable for constructing non-trivial multimode entangled quantum states of light.

We revised the conclusion in lines 308 to include the previously missing reference and to discuss future opportunities.

I hope the authors can address these literature-related concerns and clarify the technical advancements and scalability issues before I finalize my recommendation regarding publication.

In conclusion, we thank the Reviewer for the constructive and helpful comments, which prompted major revisions, including a new SI. We believe that the revised manuscript better places our work in its context and clearly explains the benefits of photonic crystal resonators (PhCRs) for scalable operation, and with exciting prospects for future work. We hope the Reviewer continues to recognize our approach as novel and interesting and, given the significant revisions we made in response to the Reviewer's comments, can recommend it positively.

Reviewer #3 (Remarks to the Author):

I co-reviewed this manuscript with one of the Reviewers who provided the listed reports. This is part of the Nature Communications initiative to facilitate training in peer review and to provide appropriate recognition for Early Career Researchers who co-review manuscripts.

We thank the Reviewer for the contribution to the constructive and helpful comments that have enabled us to substantially revise and improve our manuscript. We hope that our substantial revisions and additions satisfactorily address all comments by the Reviewer, and that the Reviewer can come to a positive recommendation of our work.

Reply to the Reviewers

Reviewer #1 (Remarks to the Author):

I appreciate that the authors have addressed the technical improvement questions I raised in my previous review. However, I remain unconvinced that this work constitutes a significant advance. In my view, the results primarily demonstrate structural variations relative to the current state of the art, rather than a substantive breakthrough.

Inferred squeezing does not amount to a direct assessment of actual squeezing, as the conclusions are confounded by multiple interdependencies. The lack of tunability ultimately places the work at a disadvantage compared to existing approaches, such as photonic molecules.

We thank the Reviewer for the careful re-assessment of our work. We understand that the Reviewer considers our approach to be at a disadvantage compared to the photonic molecule platform, including with regards to tunability of the squeezing level.

We would like to point out that tuning (i.e., reducing) the degree of squeezing is indeed simple. It can be accomplished by reducing the pump power or attenuating the generated squeezed state of light; the tunability of photonic molecules is not needed.

Moreover, while we agree that coupled ring resonators (“photonic molecules”) are a versatile and powerful architecture for nonlinear photonics, we believe that, in the context of squeezing, our platform offers several key advantages:

Robustness against temperature variations: Our approach does not require maintaining a delicate detuning between two coupled resonators. The ability to tune photonic molecules via the (differential) temperatures of the coupled microresonators also introduces sensitivity to temperature variations. This is not advantageous when aiming for robust or scalable systems. As we pointed out, for a typical thermorefractive coefficient of 10^{-5} K^{-1} and a resonance width of 100 MHz, a few millikelvin of temperature stability is required to keep differential resonance shifts below a tenth of the linewidth. Achieving this simultaneously for both resonators is challenging, especially when the pump lasers couple to only one of them; active control is needed. In contrast, our platform couples counterpropagating modes within the same resonator, where the coupling strength (mode splitting) is intrinsically stable against temperature fluctuations.

Relaxed fabrication tolerances: Our approach does not rely on fabricating two resonators with a precise FSR ratio. In photonic molecules, splitting multiple resonances requires a precise ratio between the free-spectral ranges (FSRs) of the two resonators, since only a global frequency offset can be tuned. This places stringent demands on fabrication accuracy. By contrast, our platform uses a single resonator, where the coupled counterpropagating modes are intrinsically frequency-aligned for any number of split modes.

Thus, in our view, our platform provides intrinsic advantages in operational robustness and relaxed fabrication tolerances. We sincerely thank the Reviewer for the constructive remarks throughout the review process, which have helped strengthen the manuscript, and we hope the Reviewer may reconsider the conclusion drawn from the comparison with photonic molecules.

Reviewer #2 (Remarks to the Author):

The authors have done a good job addressing the reviewers' comments. The revisions have significantly strengthened the manuscript. The expanded functionalities demonstrated here surpass prior work, but also highlight the novelty and impact of this study. I am pleased to recommend this paper for publication in its present form.

We thank the Reviewer for the careful re-assessment of our work and the constructive comments in the review process that helped us to strengthen our manuscript.